# Building an Effective Classifier for Phishing Web Pages Detection: A Quantum-Inspired Biomimetic Paradigm Suitable for Big Data Analytics of Cyber Attacks

**DOI:** 10.3390/biomimetics8020197

**Published:** 2023-05-09

**Authors:** Saad M. Darwish, Dheyauldeen A. Farhan, Adel A. Elzoghabi

**Affiliations:** 1Department of Information Technology, Institute of Graduate Studies and Research, Alexandria University, 163 Horreya Avenue, El Shatby 21526, Alexandria P.O. Box 832, Egypt; 2Department of Computer Science, Al-Maarif University College, Ramadi 31001, Iraq

**Keywords:** malicious URLs detection, cyber security, big data analytics, biomimetic algorithm, quantum-inspired computing

## Abstract

To combat malicious domains, which serve as a key platform for a wide range of attacks, domain name service (DNS) data provide rich traces of Internet activities and are a powerful resource. This paper presents new research that proposes a model for finding malicious domains by passively analyzing DNS data. The proposed model builds a real-time, accurate, middleweight, and fast classifier by combining a genetic algorithm for selecting DNS data features with a two-step quantum ant colony optimization (QABC) algorithm for classification. The modified two-step QABC classifier uses *K*-means instead of random initialization to place food sources. In order to overcome ABCs poor exploitation abilities and its convergence speed, this paper utilizes the metaheuristic QABC algorithm for global optimization problems inspired by quantum physics concepts. The use of the Hadoop framework and a hybrid machine learning approach (K-mean and QABC) to deal with the large size of uniform resource locator (URL) data is one of the main contributions of this paper. The major point is that blacklists, heavyweight classifiers (those that use more features), and lightweight classifiers (those that use fewer features and consume the features from the browser) may all be improved with the use of the suggested machine learning method. The results showed that the suggested model could work with more than 96.6% accuracy for more than 10 million query–answer pairs.

## 1. Introduction

The World Wide Web is an Internet-accessible network of websites. The web has become a platform for a broad variety of criminal activities, such as spam-advertised commerce, financial fraud (phishing), and malware dissemination (Trojan downloads). If people were warned before visiting a harmful URL, this issue may be reduced. The security community has responded by building blacklisting toolbars, appliances, and search engines [1,2]. Many harmful sites are not banned because they are too new, have never been examined, or are evaluated wrongly (e.g., cloaking). To solve this difficulty, several client-side technologies assess a website’s content or behavior as it is accessed. In addition to runtime costs, these techniques expose users to browser-based vulnerabilities [3].

Malicious URLs are used by distribution channels to spread malware online. Attackers get partial or complete system control via these linkages. This leads to readily infected target computers that may be used for cybercrimes, such as stealing passwords, spamming, phishing, and denial-of-service attacks. The system should be fast, precise, and able to identify new malicious content [3,4]. Identifying the URL, domain name, or IP address of malicious activity is crucial. One of the most promising approaches is analyzing DNS data [5].

DNS transfers human-readable domain names to IP addresses that computers may use to transport packets over the Internet. DNS analysis provides various advantages over blacklists for detecting malicious domains [5,6,7]. DNS traffic provides several features to detect malicious domain names. A huge variety of attributes and traffic data make DNS traffic an ideal choice for machine learning (ML) security efforts. Due to traces left in DNS data, researchers may detect attacks before they begin. Malicious URL detection has been addressed in several ways. According to basic ideas, these techniques may be classified into three groups [1,8]: blacklisting, heuristics, and machine learning. See [8,9,10] for more details.

Machine learning algorithms analyze URLs and their corresponding websites or webpages by extracting excellent feature representations of URLs and building a prediction model on harmful and benign URLs. Static and dynamic features are utilized. Static analysis analyses a website without executing the URL. These techniques are safer than dynamic approaches since no execution is needed. Dynamic analysis involves monitoring prospective victims’ behavior for anomalies. Dynamic analysis approaches are challenging to develop and generalize. The use of machine learning to detect malicious URLs has come a long way in the last decade, but there are still many open problems and challenges that need to be addressed. These include issues with high volume and velocity, difficulty in acquiring labels, difficulty in collecting features, feature representation, and concept drifting, in which the distribution of malicious URLs may change over time due to the evolving nature of cybercrime [4,8].

Feature extraction is a difficult operation that affects the quality of malicious URL detection. Features may be passive or active [11]. Passive (internal) data may be retrieved from DNS requests from resolvers. Active (contextual) features need further external data. Choosing a minimal feature set to lower a system’s cost and attaining a high recognition rate are the primary problems in feature selection. Two categories define these feature selection strategies. First, problem-specific solutions are devised using domain knowledge to decrease the number of features. When domain knowledge is unavailable or costly to utilize, employ the second option. In this case, meta-heuristics (e.g., genetic algorithm) are applied to select a subset “*d*” of the available “*m*” features [12,13].

Genetic algorithms (GAs) are adaptive search approaches that use inductive learning. GAs are suitable for applications where domain knowledge is hard to acquire. Choosing a suitable representation and evaluation function is key to using GAs [13,14]. Recently, the artificial bee colony (ABC) optimization technique, based on honeybee foraging behavior, has been used to benchmark classification and clustering issues [15,16]. ABC adjusts each individual’s location based on the variation process, which explores unknown portions of the search space, and the selection process, which exploits the found region [17]. Recent research has shown that the ABC algorithm is more effective and may provide better outcomes than competing metaheuristic algorithms [18]. However, the ABC algorithm’s fundamental flaw is that it performs well during exploration but poorly during exploitation. In other situations, convergence is also a problem [19]. Different variants of ABC have been offered in the literature to help establish the best balance between exploration and exploitation. Recent work by Quantum ABC (QABC) uses quantum physics ideas to build a better update equation that strikes a balance between them [20,21,22].

Cyber security experts must invent and deploy unique approaches to reduce threats in a big data environment. Traditional computer storage technologies like relational databases can’t handle malicious URL detection’s huge data difficulties. Hadoop, an open-source distributed storage technology that runs on commodity hardware, is used to store big data’s large volume, high velocity, and heterogeneous data formats [23]. Hadoop lets users develop and test distributed systems quickly. It efficiently distributes data and tasks among computers and uses CPU core parallelism [24].

### 1.1. Problem Statement and Research Motivation

There are many different types of spam, phishing, and drive-by exploits that may be hosted on a malicious URL or malicious website. This means users need to think critically about the potential consequences of visiting an unfamiliar URL. A user’s decision on whether or not to visit a certain URL may be assisted in a number of different ways. Some systems examine the content of the visited webpage and evaluate user behavior after viewing the page to get around the issue of updating and maintaining a blacklist. The user experience suffers as a result, and the runtime overhead grows as a consequence. Recently, machine learning has been used for the detection of malicious URLs by means of a number of classification-based approaches that utilize aspects of web page content and URL text. In order to more effectively identify malicious URLs, classification-based approaches have been developed. Although as defenses evolve, so do the methods used by attackers, who learn from each encounter and develop novel methods of attack. The motivation for this work is the fact that attackers often misuse specific domain features throughout this procedure. We can successfully identify and block malicious websites by building a detection algorithm on such features.

### 1.2. Contribution and Methodology

Using machine learning over URL-based features, this effort attempts to solve the challenge of identifying malicious websites. With this, we want to strike a balance between the problems experienced by blacklists (due to their reliance on human updates that cannot keep up with newly introduced harmful sites) and client-side solutions (which are difficult to deploy on a large scale because of their high overhead). Building a reliable, scalable, and flexible system is a formidable obstacle. The work presented in this paper is an extension of the work introduced in Ref. [25], where quantum mechanics was used in the ABC optimization algorithm to make it easier for particles to get out of the local optimum so that the malicious URL could be detected more accurate. The following are some of the work’s contributions towards this goal:-This work contributes by demonstrating that the improvements obtained by utilizing a specialized learner are outweighed by the advantages made by extracting more meaningful features from the URL data, allowing for improved classification. As a result, GA is adapted into a bio-inspired feature reduction approach to focus on the most pertinent features (optimal URL data features) for constructing a powerful and effective learning model.-A two-step quantum artificial bee colony (2-step QABC) technique is adapted for effective malicious URL clustering. The *K*-means method is used to determine the starting locations of food sources in the modified 2-step QABC algorithm, rather than utilizing randomization. In addition, a refined solution search equation inspired by particle social behavior was utilized to focus on the most successful places to look.-Real-world DNS traffic may be much higher than that in published datasets. So, malicious website detection methods must be scalable. Further, certain methods need big data for detection algorithm training and fine-tuning. The proposed strategy takes advantage of the Apache Hadoop distributed computing technology to solve this issue.

This article continues as follows: Section 2 reviews big data malicious URL detection strategies. Section 3 provides an outline of the recommended strategy. Section 4 tests the suggested approach and reviews the results. Section 5 summarizes the presented work and recommends next directions.

## 2. State of the Art

Figure 1 depicts URL classification. It reflects URL-related features, datasets, learning methodologies, models, and attack kinds [26]. URLs have many features. Malicious URLs are further categorized by attack kinds, including spam, phishing, and malware. This section discusses relevant work in URL classification utilizing machine learning techniques, statistical methods in related applications, and other approaches to URL classification. Because the recommended approach is dependent on passive domain name-based URL elements, greater focus is given to surveying-related work that integrates such aspects.

In [27], the authors developed a method to automatically categories URLs as malicious or benign using lexical and host-based features. Their technique is complementary to a blacklist that cannot forecast the status of previously uncovered URLs and algorithms that evaluate site content and behavior by visiting potentially harmful sites. They also demonstrated that with suitable classifiers, it is possible to automatically filter through complete feature sets (without domain knowledge) and pick the best predictive features for categorization. They obtained 95–99% accuracy in classifying harmful websites from URLs, with relatively few false positives. However, it is unclear how to expand this strategy to accommodate millions of evolving URL data features [28].

In [29], the authors presented a machine-learning-based phishing classifier. By automatically updating their blacklist using a classifier, they lowered phishing page availability. Even with a great classifier and a strong system, their blacklist method puts them behind phishers. A phishing page can only be identified after it has been online for some time. Despite noise in the training data, their classifier builds a strong model for recognizing phishing sites and accurately identifies 90% of them weeks after training. Their algorithm solely classifies user-submitted web pages and Gmail spam URLs [30]. In [31], the authors offered a content-based method for identifying phishing websites. This strategy depended on a simple categorization model that was less accurate than sophisticated classification models [32].

The work presented in [33] presented a bag of words for URL classification. Their approach used maximum-entropy machine learning. To train their models rapidly, they simply employ lexical features, analyzing the URL string and disregarding page content and host information. They employ a bag-of-words representation of URL tokens with location annotations (e.g., hostname, path, etc.). They employ successive N-grams of tokens, sorted but non-consecutive bigrams, and URL lengths. Lexical characteristics can match page content features with 95% accuracy. Concurrently, Guan et al. [34] looked into instant messaging (IM) for categorizing URLs. Although they employ URL-based capabilities, they also use IM-specific aspects, including message timing and content. URL-based characteristics include domain age (WHOIS domain name), Google rank, and lexical features. They utilize an ad-hoc linear classifier where the weight of each attribute is proportional to the number of benign and malicious samples that have it. This algorithm requires additional URL message samples to be more accurate and convincing.

In [35], the authors studied ways to identify fraudulent websites using lexical and host-based URL features. They created a real-time system to capture URL features and paired it with a real-time feed of labelled URLs from a big webmail provider. With these features and labels, they can train an online classifier to identify malicious websites with 99% accuracy on a balanced dataset. Watkins et al. [36] developed a malicious website detection strategy based on big data. One solution to eliminate the big data issue in cybersecurity is to ignore most of an enterprise’s non-malicious network data and concentrate on suspicious or malicious network activity. Their solution employed basic clustering and a dataset augmented with known malicious domains to filter out non-suspicious network traffic.

Bilge et al. [37] are building an exposure system to identify such malicious domains in real-time using 15 unique features in four categories. They ran a controlled experiment with billions of DNS queries. In this case, the data collector collects the network’s DNS traffic. Then, it assigns features and domains in the database with DNS traffic features. The collection of malicious and benign domains acts independently and simultaneously with the data collector module. From multiple sources, it gathers benign and malicious domains. Daily use, query count, and target IP addresses of each domain are provided on the service’s website; the results are positive. However, the authors don’t train for malicious domains that are conceptually unknown and have never been seen in the field by malware analysts, tools, or specialists. Newer research on identifying fraudulent websites with the use of DNS properties may be found in [30,38,39,40,41,42,43,44,45,46,47].

According to the review, earlier research focused on: (1) blacklisting, which cannot anticipate the status of previously unknown URLs or systems based on site content and behavior, which requires accessing potentially risky sites; and (2) not addressing the selection of an optimal feature set from retrieved features. With proper classifiers, it is possible to automatically shift through large feature sets (without domain knowledge) and discover the best predictive features for classification. To our knowledge, quantum-based evolutionary feature selection techniques for malicious URL detection systems that rely on a large number of training samples (big data environment) have received little attention. Most bio-inspired solutions for malicious URL identification use two or more algorithms to improve exploration and exploitation. Recent work by quantum evolution uses quantum physics ideas to build a better update equation that strikes a balance between them. The following section details a concept that combines the Hadoop framework for big data with a two-step artificial bee colony algorithm for URL categorization.

## 3. Proposed Malicious URL Detection Model

In this part, we have a look at the quantum-based, bio-inspired methodology presented for passive DNS detection of malicious URLs in a large data environment.

### 3.1. Problem Formulation

In order to identify one of two possible classes, malicious or benign, the detection of malicious URLs might be posed as a binary classification issue. To be more precise, let’s say we have a training set of T URLs, represented by the pairs $\left\{\left(u_{1},y_{1}\right),\ldots ,\left(u_{T},y_{T}\right)\right\}$, where each $u_{t}$ for t=1,…,T represents a URL in the training data, and each $y_{t}=1$ indicates a malicious URL and each $y_{t}=-1$ represents a benign URL. There are two main issues with automating the identification of malicious URLs [11,12,30,38,46]:-Feature representation: Determine the best feature representation to extract, $u_{t}\to x_{t}$ in which the URL is represented by the *d*-dimensional feature vector $x_{t}{\in R}^{d}$.-Machine learning: Learning a prediction function *f*, $R^{d}\to R,$ that correctly predicts the class assignment for every instance of a URL based on the presentation of its features.

For malicious URL identification, machine learning aims to maximize prediction accuracy in binary classification tasks. Both folds above are required for this. While the first portion of feature representation is based on domain knowledge and heuristics, the second half focuses on training the classification model through data-driven optimization [2,5,10]. Malicious URL detection uses several features. Many feature categories use bag of words (occurrence) features, resulting in millions of features. In real life, as the number of URLs to be analyzed grows, so does the number of features. Using so many features to learn prediction models has two problems [11,12,38,47,48,49]: (1) It is computationally expensive, as training and test cycles are long due to the various mathematical operations and features collected and preprocessed. (2) Malicious URL detection frequently has more features than training cases. Overfitting may come from optimizing such models.

### 3.2. Methodology

Firewalls, key components for secured network infrastructures, are faced with two different kinds of challenges: First, they must be fast enough to classify network packets at line speed. Second, their packet processing capabilities should be versatile in order to support complex filtering policies. Unfortunately, most existing classification systems do not qualify equally well for both requirements; systems built on special-purpose hardware are fast but limited in their filtering functionality. In contrast, software filters provide powerful matching semantics but struggle to meet line speed. The key challenge in such a design arises from the dependencies between classification rules due to their relative priorities within the rule set: complex rules requiring software-based processing may be interleaved at arbitrary positions between those where hardware processing is feasible [4,6,8,46]. The superiority of the proposed approach on the physical platform (Firewall) lies in the fact that it is effective against modern attacks, cannot be circumvented with spoofing, cannot make decisions based on application or authentication, and allows for a huge, easily managed rule list.

In this work, a genetic algorithm was utilized in order to select the most representative features while maintaining malicious URL classification accuracy on par with state-of-the-art models that employ hundreds or thousands of features. This would make it possible for embedded programmers to search for features matching malicious URL behavior quickly. To train and optimize the learning prediction algorithm, this model needs big datasets. As a distributed computing platform, Apache Hadoop solves this challenge. Figure 2 depicts the key components of the recommended prediction model and how they’re related. The next subsections outline its phases.

#### 3.2.1. Training Phase

##### Step 1: Passive DNS Dataset

Passively collecting DNS data involves using DNS server logs to acquire genuine DNS requests and responses. Passive DNS data is more complete than active data. Passively captured DNS data are more representative and revealing in terms of features and statistics that may be used to detect malicious activities [37,42]. Because passive DNS data are linked to user activity, they can be used to identify malicious domains based on user behavior, e.g., temporal statistics of user queries. The experiments are conducted using a benchmark DNS passive dataset (https://www.circl.lu/services/passive-dns/, accessed on 1 May 2022) that is comprised of captured passive DNS data, which are answers (e.g., IP address, time to live (TTL), record counts) from authoritative DNS servers given domain name queries from the browsers of users. A 184 million-row dataset is extracted. Each row of the dataset has query-answer metadata, i.e., a query from the browser and an answer from the authoritative DNS server. Here, DNS domain name, TTL, and DNS response-based features are used to collect domain activities. According to [36], DNS domain name-based features should assist in distinguishing readable from unreadable names. Computer-generated domain names are likely indicators of malicious domains. TTL-based features may assist in distinguishing between high-availability (high TTL values indicate malicious domains) and low-availability websites. The authoritative DNS server provides DNS-answer-based functionality. The authors in [37] employed largely IP-based computations, but because our dataset was anonymized, this was not feasible, so IP addresses were removed from the dataset. Instead, we used features that focused on capturing the returned data’s attributes’ behavior.

##### Step 2: Data Preprocessing and Management

The big data issues offered by malicious URL identification are insurmountable for conventional computing storage systems like relational databases. In addition, certain methods need big datasets for training and fine-tuning detection algorithms [50]. Some academics have suggested utilizing distributed computing frameworks like Apache Hadoop to remedy this issue. Hadoop is a software infrastructure for shared data storage and processing. In order to store a large file over several nodes, it first splits it up into smaller chunks. By breaking down a large task into smaller ones, it is able to execute distributed processing. These processes are distributed throughout the cluster of computers in parallel [50]. For further information addressing the fundamentals of Hadoop MapReduce, see [51,52].

Data cleaning, integration, transformation, and reduction are all examples of preprocessing operations that assist in turning raw data into a processed and understandable format; details may be found in [53,54]. Data cleaning is the basic phase of data preparation, identifying defective data and incorrect information. The cleaning stage modifies or deletes incomplete, erroneous, inconsistent, and irrelevant data. Data cleaning can be performed on Hadoop projects using Apache Hive. Hive supports custom user-defined functions (UDF) for tasks, such as data cleansing and filtering. Hive UDFs can be defined according to programmers’ requirements [51]. Hive operates on the cluster’s server-side [52]. Each Hadoop cluster does this process separately and in parallel.

##### Step 3: Feature Extraction

Feature extraction is a dimensionality reduction procedure that reduces a collection of raw variables to more manageable groupings (features) for processing [6,8,12,13]. The success of a machine learning model depends on the quality of the training data and feature representation. Two phases comprise feature representation: (1) Feature collection is an engineering-focused phase involving the collection of important URL information. (2) In feature preprocessing, unstructured URL information is processed and turned into a numerical vector for machine learning techniques. Researchers offer many factors for detecting malicious URLs. They classified these aspects as black-list, URL-based lexical, host-based, and content-based (context, popularity, etc.) [48,49]. All have their pros and cons; although some are quite informative, they may be expensive. Similarly, different features have distinct preprocessing issues and security considerations. The blacklist, context, and popularity features need extra dependencies and have a greater collection overhead than the others. This also means that for a live system (i.e., real-time malicious URL detection), features with a high collecting time may not be possible.

The blacklist collection time might be significant if the external dependency is requested during runtime. If the complete blacklist can be kept locally, the collection overhead is modest. The collection of lexical features is efficient since they are URL extensions. Host-based functions are time-consuming. Content features require downloading the web page, which slows collection. Lexical characteristics have great dimensionality (and so do unstructured host features and content features). Here are the extracted features from the passive DNS dataset [36,37]:-Type 1: DNS answer-based features: A domain’s DNS answer consists of DNS records.-Type 2: TTL value-based features: Every DNS record has a TTL that indicates how long a domain’s answer should be cached. Both DNS clients and name servers may profit from DNS caching if the TTL is between 1 and 5 days. High-availability systems reduce hostname TTL and employ round-robin DNS.-Type 3: DNS domain name-based features: DNS provides human-readable names to people who cannot remember server IP addresses. Good internet services pick easy-to-remember domain names. Malicious individuals do not care about easy-to-remember domain names. This feature represents QNAME query name in which the target domain-name is presented as a sequence of labels, each label consisting of a length octet followed by that number of octets. The domain name ends with the zero-length octet for the null label of the root. This DNS server looks for resource records (RRs) that match the specified Qtype and Qclass. If it does not have a match, this server can point to a DNS server that may have a match instead. QTYPE is a two-octet code which specifies the type of the query, e.g., host addresses. QCLASS is a two-octet code that specifies the class of the query, e.g., internet addresses. Table 1 shows a record for the used dataset (17 feature vector per URL) [36,37,38,39,40,41,42,43,44,45].

##### Step 4: Feature Selection Using Genetic Algorithm

The challenge of feature subset selection is selecting a subset of a dataset’s original features such that an induction method operating on just the chosen features generates the most accurate prediction model. It is important to choose a selection of relevant, nonredundant features. Heuristic search narrows a wide search space of solutions to find a good one. We may think of *d* as the required number of features to be included in the subset X,X⊆Y, and Y as the original set of features with cardinality *n*. Let $J\left(X\right)$ stand in as the set X’s feature selection criteria function. Let’s assume a larger value of *J* indicates a more robust collection of features to work with. If we want to maximize $J\left(.\right)$, then one suitable criteria function is $\left(1-p_{e}\right)$, where $p_{e}$ the error probability is. Features are chosen based on the classifier used, the size of the training and testing datasets, and the chance of error [55]. Finding a subset X⊆Y such that $\left|X\right|=d$ is the formal definition of the feature selection issue.
(1)$$J\left(X\right)=\underset{Z\subseteq Y,\left|Z\right|=d}{\max}J(Z)$$

An exhaustive search is not feasible for even low values of *n* since it would entail inspecting every conceivable d-subset of the feature set *Y*, for which there are nd possibilities. Any potential ordering of the error probabilities of each of the $2^{n}$ feature subsets is feasible. Hence, there is no exhaustive sequential feature selection technique that is guaranteed to provide the best subset. Forward selection and backward selection are two types of feature subset search techniques; see [56] for more details. The basic feature selection procedure is shown in Algorithm 1 [57].
**Algorithm 1:** Feature Selection AlgorithmInput: 
S—Data sample with features $X,\left|X\right|=d$

J—Evaluation measure to be maximized 
GS—Successor generation operator Output: Solution Begin: Solution: = (weighted) feature subset                    L≔start_point(X);
Solution: = $\left\{best\;of\;L\;according\;to\;J\right\};$
Repeat:                 $L≔search\_strategy(L,GS\left(J\right),X);$
                $X^{\prime }≔\left\{best\;of\;L\;according\;to\;J\right\};$

$if\;J\left(X^{\prime }\right)\geq J\left(Solution\right)\;or\;\left(J\left(X^{\prime }\right)=J\left(Solution\right)\;and\;\left|X^{\prime }\right|<\left|Solution\right|\right)then$
                    $Solution=X^{\prime };$
Until Stop (*J*, *L*).

In this case, we chose the genetic algorithm as a feature selector because of its ease of use and robustness in the face of noisy data. Algorithm 2 shows the pseudo code of the genetic algorithm. A GA-feature selection optimization problem instance may be formally characterized as a four-tuple (*R*, *Q*, *T*, *f*) defined as [57,58,59,60]:

-*R* is the solution space; each chromosome contains 17 feature vector per URL (see Table 1). *R* is *n*× matrix, where *n* is the number of URL samples. Each bit is a gene representing the vector’s feature.-*Q* is the feasibility predicate (different operators: selection, crossover, and mutation). Crossover refers to the process by which genes from one parent are swapped with those from another parent in order to create a hybrid child. Simple single-point crossovers are used here. A uniform mutation is used to avoid slipping into a locally optimal solution. The selection operator keeps the best-fitting chromosome from one generation and chooses specified numbers of parent chromosomes. Tournament selection is popular in genetic algorithms because of its efficiency and simplicity [61].-ζ represents the collection of viable options (new-generation populations). Following these iterations, the most optimal chromosome will be used to symbolize the URL feature vector by including a selection of key components. According to the identification success rate, this vector will explicitly indicate the best possible feature combination [62].-*f* is the fitness function. The fittest individual will become the operator’s companion. The fitness function is based on the difference between the actual URL’s categorization and its calculated one. *Acc* is given by [11]:


(2)
Acc=(TP+TN)/(P+N)


**Algorithm 2:** Genetic Algorithm Pseudo Code

*t = 0*
Generate Initial Population [*R*(*t*)];Evaluate Population [*R*(*t*)];**While** not termination **do***R’*(*t*)= Variation [*R*(*t*)];Evaluate population [*R’*(*t*)];*R*(*t* + 1) = Apply GA Operators [*R’*(*t*) ∪ *Q*];*t* = *t* + 1
**End while**



Accuracy (*Acc*) is the ratio of the correctly identified domains to the whole size of the test set. The higher the value is, the better $\left(Acc\epsilon \left[0,1\right]\right).$ True positive (*TP*) is the correctly identified malicious domains. True negative (*TN*) is the correctly identified benign domains, *P* is the total number of malicious domains, and *N* is the total number of benign domains. Based on the results, a 10-element feature vector is optimal for use, allowing the proposed classifier to achieve the lowest error rate. Remember that the GA’s non-deterministic nature implies that the optimal feature vector’s elements may vary with each program run. In general, the use of the genetic algorithm results in the reduction of the feature vector size 1 × 17 to a feature vector of size 1 × 10. So, the number of features (optimized features) used for each URL was reduced to 58% of the total number of features (all features).

##### Step 5: Quantum Artificial Bee Colony (QABC) Classifier

In the last stage, a quantum artificial bee colony classifies URLs as malicious or benign based on the training dataset’s best feature vector. This study uses a two-step QABC method to enhance the ABC algorithm for clustering [63] issues by employing the *K*-means algorithm (see Algorithm 3). The combination of QABC and *K*-means solves the issue of malicious URL categorization. The classifier pseudo-code (see Algorithm 4) comprises four phases: initialization, employed bee, onlooker bee, abandoned food source, and scout bee phases. In the initialization step, food supply locations for employed bees are estimated using the *K*-means method. The employed-bees phase exploits nearby food sources. The new food source position relies on the previous food source location, the employed bee position, and a [0, 1] random variable. Then, a probability is generated for each food source based on its quality. The observer bee phase uses a quantum behavior-based searching mechanism to pinpoint the position of the bee swarm to determine food quality. If this is good enough, the current food supply is chosen; otherwise, onlooker bees search for a new food source. If the solution isn’t updated in the Scout Bee or abandoned food source phase, the current food source is abandoned. Current bees work as scouts to find a new food supply. This method continues until it reaches a predetermined threshold or limit iterations [20,21,22,63]. The recommended model’s fitness function is sensitivity. Sensitivity is the ratio of true positives to true positives plus false negatives. False negatives are malicious domains that were wrongly detected as non-harmful. True positives are accurately discovered malicious domains.
(3)$$Sensitivity=\frac{True\;Positive}{True\;positive+False\;negative}$$

While in Newtonian physics, a particle’s path may be predicted with reasonable accuracy, this is not the case in quantum mechanics. Since Heisenberg’s uncertainty principle prevents us from knowing both the *x* (position) and *v* (velocity) of a particle at the same time, the word “trajectory” is useless in quantum physics. Therefore, the ABC method will operate differently if the particles making up the system exhibit quantum characteristics [20,21,22]. Each bee in a quantum model of ABC is a particle with a state represented by a wave function rather than a location and velocity. The dynamic behavior of the bee is distinct from the bee in the normal ABC method in that it is not possible to concurrently compute exact values for *x* and *v*. The bees’ position probability distribution is ${\left|\Psi \left(x,t\right)\right|}^{2}$. The update function $V\left(x\right)$ may be obtained as follows:(4)$$V\left(x\right)=-\lambda \delta (x)$$
where λ is a positive value and δ(x) is Dirac delta function that simplifies calculations required for the studies of electron motion and propagation. A particle’s wave function $\Psi \left(x\right)$ in delta potential is as:(5)$$\Psi \left(x\right)=\sqrt{\frac{m\lambda }{\hbar^{2}}}(e^{-\frac{m\lambda \left|x\right|}{\hbar^{2}}})$$
where ℏ is the reduced Plank constant, the quantization of angular momentum, m is the mass of the particle and *e* is the energy of the particle. Based on Eberhard’s convergence study for PSOs, we may set the potential’s base at the position specified by:(6)$$p_{j}=\frac{r_{1}p_{ij}+r_{2}p_{gj}}{r_{1}+r_{2}},j=1,2,\ldots SN/2$$
where $p_{ij}$ is the *i*th bee local best position in the *j*th dimension of the hyperspace and $p_{gj}$ is the value of the *j*th dimension of the bees’ global best position; $r_{1}$ and $r_{2}$ are random variables in the range (0, 1].
(7)$$V\left(z\right)=-\lambda \delta \left(x-p\right)=-\lambda \delta (z)$$
(8)$$\Psi \left(z\right)=\sqrt{\frac{m\lambda }{\hbar^{2}}}(e^{-\frac{m\lambda \left|z\right|}{\hbar^{2}}})$$
where z=x−p. As a result, we can calculate the probability of finding a particle *Q*(*z*) at any given location:$$Since\;Q(z)={\left|\Psi (z)\right|}^{2,}$$
(9)$$∴Q(z)=\frac{m\lambda }{\hbar^{2}}exp(-2\frac{m\lambda \left|z\right|}{\hbar^{2}})$$

The *F*(*z*) cumulative distribution function is defined as:(10)$$F(z)=\int_{-\infty }^{z}\frac{m\lambda }{\hbar^{2}}exp(-2\frac{m\lambda \left|z\right|}{\hbar^{2}})=e^{\frac{-2\hbar^{2}\left|z\right|}{m\lambda }}$$
(11)$$V=P\pm \frac{\hbar^{2}}{2m\lambda }ln(\frac{1}{u})$$

Herein, *u* = rand (0, 1), and let $c=\frac{\hbar^{2}}{2m\lambda }$ then
(12)$$v_{i}=P_{i}\pm c_{i}ln(\frac{1}{u})$$

We may define *MB* as the mean of the best *SN*/2 positions. *SN* is the size of food sources.
(13)$$MB=\frac{2}{n}\sum_{i=1}^{SN/2}p_{i}$$

Let $\vec{\mu }=[\mu_{1},\mu_{2},...,\mu_{\left(\frac{SN}{2}\right)}]$, and $\vec{\mu }$ is a tuning factor vector that is listed in decreasing order. To alter the distribution’s variance so that the algorithm may successfully complete the optimization objective, let:(14)$$c_{i}=\mu_{i}*(\left|MB-x_{i}\right|)$$
(15)$$v_{i}=\left\{\begin{array}{c} p_{i}+\mu_{i}*(\left|MB-x_{i}\right|ln(\frac{1}{u})\;for\;k\geq 0.5\\ p_{i}-\mu_{i}*(\left|MB-x_{i}\right|ln(\frac{1}{u})\;for\;k<0.5 \end{array}\right.$$
where *k* is a random number between zero and one, uniformly distributed.
**Algorithm 3: ***K*-means clustering **Input**: *K* (the number of clusters); *K* = 2 (malicious or benign) *D* dataset contains best feature values $f_{best,i}$ for each *URL_i_*
**Output**: $Cluster\;S_{j}$
**Begin**
Arbitrary choose *K* objects from *D* as the initial cluster centers; **Repeat**
-(re) assign each object to the cluster to which the object is the most similar, based on the mean value of the objects in the cluster;-Update the cluster means, i.e., calculate the mean value of the objects for each cluster
**Until no change;**


**Algorithm 4:** Two-step QABC clustering Input: *D* dataset contains best feature values $f_{best,i}$ for each ${URL}_{i}$
Fitness Function “Sensitivity” Output: Best solution of final cluster center ($C_{best,j}$) *j* = 1, 2 **Begin**
**Initialization phase.**
-Important QABC settings are determined by the user, including particle population size, optimization variable bounds, modification rate (*MR*), tuning vector ($\vec{\mu }$), and stopping criterion (*t*_max).-Use a uniform probability distribution function to seed an initial population (array) of food sources with placements across the n-dimensional space.
**For**
*i* = 1:*SN*/* *SN* is the total number of food sources (number of clusters) */ Initialize the food source within the boundary of given dataset in random order; Apply the *K*-means algorithm Send the employed bees to the food sources; /* Computed centers */ **End For**
Iteration = 0; **Do While** (the termination condition is not met) **For** (each employed bee)/* Employed bee’s phase */ For *i* = 1:*SN*
-Update the positions of the employed bees using $v_{ij}=x_{ij}+\emptyset_{ij}\left(x_{ij}-x_{kj}\right),\emptyset_{ij}\in [-1,1]$, then evaluate their fitness using Equation (3)-A greedy selection is made between the old and new food source and keep the best one;
**End For**
**For**
*i* = 1:*SN*
Compute the probability value associated with each food source.   **End For**
**For**
*i* = 1:*SN* /* Start the onlooker bees phase*/   **If** (rand ( ) < *Pr_i_*) /* *Pr_i_* the probability associated with ith food source */
-Generate a new position for each onlooker using Equation (15) then evaluate the fitness-A greedy selection is made between the old and new food source and keep the best one;
**Else**
*i* = *i* + 1; **End If**
**End For**
**If** (any employed bee becomes scout bee)/* Scout bee’s phase */Send the scout bee to a randomly produced to food source;**End if**
Memorize the best solution achieved Iteration = iteration + 1 **End While**Obtain final cluster Center**End**

#### 3.2.2. Testing Phase

Machine learning aims to predict test data. Training data is used to fit and test the model. Models are created to predict test-set outcomes. Given an unknown URL, the model extracts its feature vector by following the indices of the best feature vector obtained during training. This extracted feature vector is then categorized based on its resemblance to the final cluster centers created during the training phase. In summary, the recommended model uses a two-step ABC approach that uses *K*-means to determine initial cluster centers. Excellent exploration and exploitation in the classifier’s solution search leads to good convergence. The convergence speed of the ABC method slows as problem dimensions rise. To solve this difficulty, the recommended model reduces the number of features. The complete pseudo-code of the suggested model is shown in Algorithm 5.
**Algorithm 5:** the Suggested Clustering Model  **Input:**
*X_i_*: *Passive DNS Dataset*
*GA parameters configuration*
*ABC classifier parameters configuration*
**Output**: URL classifier (malicious, benign) **Begin**     1. Initially load dataset in Hadoop     2. Apply preprocessing in multiple parallel operations     3. **For each** MapReduce partitions     4. Apply feature extraction procedure     5. **End for**
     6. Aggregate selected features     7. Call algorithm 1 for feature selection     8. Call algorithm 4 for two-step QABC clustering     9. Classify unknown URL according to the trained classifier
**End**

The primary benefits of the provided approach are: (1) reducing computing complexity by using GA to pick the ideal feature without affecting detection accuracy. (2) Improving malicious URL detection using a two-step ABC classifier. The classifier mixes *K*-means and a bio-inspired ABC classifier that handles the tradeoff between exploration and exploitation well. Exploration is the capacity to analyze prospective solutions that are not neighbors of the current answer (or solutions). This procedure helps escape a local optimum. Exploitation occurs when a search is performed in the vicinity of the present solution (or solutions). It may be implemented as a local search.

## 4. Results and Discussions

This section analyses the model’s efficiency. Experiments were performed to test the model’s resilience. The experiments employ a benchmark dataset of collected passive DNS data (https://www.circl.lu/services/passive-dns/, accessed on 1 May 2022), which are replies (e.g., IP address, TTL, record counts) from authoritative DNS servers given domain name requests from user browsers. CIRCL Passive DNS is a database storing historical DNS records from various resources including malware analysis or partners. The DNS historical data are indexed, which makes it searchable for incident handlers, security analysts, or researchers. In this case, we’re working with a big 184 million-row dataset. Each row in the datasets included information about a query and its corresponding response (i.e., a query from the browser and an answer from the authoritative DNS server). In this case, we make use of the DNS domain name, TTL, and DNS response-based characteristics to guarantee that a complete range of domain behaviors is recorded. The machine learning algorithm was trained on varied ratios of malicious and secure websites; 70% of the labelled pages were for training, 30% for testing. Testing and training supervised machine learning models use the same ratio of malicious to safe websites.

Java was used to build all of the components of the proposed model. With the following configurations, the system was implemented on an HP ProLiant DL180 Gen9 server. Hardware configuration: 64-bit OS, 32 GB RAM, 12× Intel Xeon E5-2620v3 processors at 2.4 GHz, and an SSD. The server has three 1 TB hard drives in a RAID 5 array and runs the 64-bit enterprise edition of Microsoft Server 2008 R2. The effectiveness and functionality of the recommended supervised algorithms are evaluated using a confusion matrix. The suggested machine learning approaches are structured in a way that prioritizes identifying malicious websites, often known as true positives [11].
(16)$$\mathrm{False}\;\mathrm{positive}\;\mathrm{rates}=\frac{FP}{N}$$
(17)$$\mathrm{Precision}=\frac{TP}{TP+FP}$$
(18)$$\mathrm{Accuracy}=\frac{TP+TN}{TP+TN+FP+FN}$$
(19)$${TN}_{rate}=\frac{TN}{TN+FP}$$
(20)$$\mathrm{Recall}={TP}_{rate}=\frac{TP}{TP+FN}$$

In computer science, the analysis of algorithms is the process of finding the computational complexity of algorithms, i.e., the amount of time, storage, or other resources needed to execute them. Usually, this involves determining a function that relates the size of an algorithm’s input to the number of steps it takes (its time complexity) or the number of storage locations it uses (its space complexity). Time complexity is commonly estimated by counting the number of elementary operations performed by the algorithm, where an elementary operation takes a fixed amount of time to perform. As the prototype of the proposed model was built using off-the-shelf software that contains many functions that call each other, it becomes theoretically difficult to calculate time and space complexities. Time complexity is a complete theoretical concept related to algorithms, while running time is the time a code would take to run. Run-time analysis was utilized as an indicator of the different approaches’ computational complexity. An algorithm is said to be efficient when this function’s values are small or grow slowly compared to the size of the input [64]. On average, the proposed technique requires around one third as much time to run in the testing phase as it does in the training phase to accomplish classification.

### 4.1. Experiment 1: (The Significance of Features Selection)

This experiment implements the recommended quantum- inspired malicious URL detection model using both a complete feature vector and an optimized feature vector to compare detection accuracy and time. Table 2 shows that compared to the whole feature vector, the application of optimum features improves detection accuracy by approximately 2% in terms of true positive (*TP*), true negative (*TN*), false positive (*FP*), and false negative (*FN*). Although this boost is modest, the time it takes to test 1000 URLs is reduced from around a 2300 to 760 milliseconds (testing phase-online). This feature vector, as expected, produces rising detection accuracy as the suggested model attempts to choose the most significant features that comprise the URL properties that may identify websites as malicious or benign. The fact that the feature selection module can get rid of unnecessary features (highly correlated features) and features that cause mislabeling (based on the fitness function) may explain these results.

### 4.2. Experiment 2: (Classifiers Evaluation)

Since there are many supervised classification techniques, this set of experiments evaluates a sample of the obtained dataset using Weka [65,66] as well as ABC, two-step ABC classifiers, and the proposed QABC. Three classifiers were examined, encompassing tree-based (random forest, C4.5) and function-based (SVM). Categorization was performed without parameter adjustment by ten-fold cross-validation to choose the most promising strategy. Table 3 displays the results for each classifier in terms of accuracy, true positives, and true negatives. SVM has the lowest accuracy (87%) but the highest efficiency (93% success rate) for detecting genuine URLs compared to the other evaluated classifiers. Similar performance (approximately 95%) is shown with tree-based classifiers, e.g., random forest, but with a greater disparity between true positives and false negatives.

The ABC classifier performs similarly to the random forest. The quantum ABC classifier classifies 98% of URLs correctly. The ABC classifier takes a lengthy time due to its stochastic nature. Bees and food sources are initially selected randomly, which makes optimization more difficult. The low convergence is a consequence of the solution search equation of ABC being excellent at exploration but not exploitation. In addition, it is observed that the ABC method slows down in its convergence as the problem’s dimensions get larger. The two-step ABC method uses the *K*-means algorithm to determine the beginning placements of food sources rather than a random starting point. Therefore, it provides more precise categorization results. The QABC classifier is based on the same idea as the two-step ABC technique. However, it employs the metaheuristic searching algorithm for global optimization problems motivated by principles from quantum physics in order to overcome ABC’s poor exploitation abilities and its convergence speed. The suggested model takes more time to classify URLs than competing approaches, as indicated in Table 2, but it also provides higher accuracy.

### 4.3. Experiment 3: (Tuning False Positives and Negatives)

False positives are more tolerable than false negatives for identifying phishing URLs. False positive URLs require visitors to be extra careful while loading the URL and manually authenticate the webpage’s legitimacy before providing sensitive personal information. False negatives may give users a false sense of security, and they may provide personal information to a phishing website. Instead of decreasing the total error rate, users may choose to tune the decision threshold to reduce false negatives and increase false positives, or vice versa. A receiver operating characteristic (ROC) plot shows how well a binary classifier can identify a problem when the threshold for making a distinction between classes changes. The ROC graph for the proposed two-step QABC classifier over a sample of the whole data set with the best feature vector is shown in Figure 3. This graph indicates the trade-off between the false negative and false positive rates. The highlighted portion of the chart reveals that a false negative rate of 3.16% may be attained by tuning the false positives to 0.15%. However, we may reduce the false negative rate to 1.05% if we accept a somewhat higher false positive rate of 0.4%. The proposed model’s accuracy in terms of false positives and negatives may be attributed to its investigation of a hybrid machine learning approach (K-mean and ABC) that utilized selected discriminative features and quantum searching mechanisms to increase diversity among populations.

### 4.4. Experiment 4: (Concept Drift)

Phishing strategies and URL architectures change as attackers find new ways to bypass filters. As phishing URLs change, the classifier’s trained model must improve. Retraining algorithms with new features is vital for adjusting to malicious URLs and their properties. The passive DNS data collection is separated randomly into 10 batches, each comprising 1,500,000 benign and malicious URLs. In our case, the OperateBatchDomain function was used to submit a task for adding domain names or DNS records in batches. In order to verify the classification error rates of the proposed model, it was trained using each batch as a distinct training set. Table 4 displays the percentage of incorrect classifications made by the proposed classifier after being trained on a variety of batches.

The results show that the overall error rate for all patches is between 2% and 2.30%. Because of this, the proposed model shows that its error rate is consistent across different datasets. These outcomes may be explained by the fact that the recommended model is constructed using an optimal feature vector that is able to discriminate between safe and harmful URLs. With this feature vector, we may reduce the degree of similarity across classes while increasing it within them. Distances between data points and the cluster centers are displayed by inter-class clusters, while intra-class distances are displayed by the clusters themselves.

### 4.5. Experiment 5: (Performance of Different Metaheuristics-Based Feature Selection Algorithms)

Feature selection is the process of selecting the best feature among a huge number of features in a dataset. However, the problem in feature selection is to select a subset that performs better under some classifier. Feature selection is frequently used in machine learning for processing high-dimensional data. It reduces the number of features in the dataset and makes the classification process easier. This procedure can not only reduce irrelevant features, but, in some cases, also increase classification performance due to the finite sample size. Meta-heuristic algorithms are widely adopted for feature selection due to their enhanced searching ability. Though current state-of-the-art systems have demonstrated impressive performance, there is still no consensus on the optimum feature selection algorithm for the task of phishing web page detection.

In this experiment, we investigate the performance of three metaheuristic feature selection algorithms: ant colony optimization (ACO), binary bat (BB), and binary grey wolf (BGW). In this case, we swap between different feature selection modules in step 4 and compare the results to those obtained by the genetic algorithm (GA) that is utilized by the proposed model for feature selections for the same task. For implementation, each module was embedded into the main model as a black box with its default parameters. See [67,68,69,70,71,72,73] for more details about the mentioned metaheuristic optimization techniques and their default parameters. As revealed in Table 5, both GA and ACO have gained better classification accuracy than other algorithms. However, utilizing GA achieves 50% feature reduction as compared with ACO, which achieves only 40%. There is no significant difference between them in the accuracy of the classification, but the use of GA for feature selection reduces the computational complexity by 20% on average. In general, one of the factors that influences the increment of time and iteration is the population being initialized. The way ACO initializes the population by using state transition rules is more efficient compared to GA, which is based on random approaches.

### 4.6. Limitation

-Malicious websites are influenced by the selected URL features. They may be altered by several factors. Classification accuracy is the only metric in this work.-As such, the proposed supervised machine learning model cannot evaluate the harmful potential of any given domain. If a domain isn’t being resolved by any hosts, for instance, it won’t be recorded in the passive DNS database, and hence won’t be considered in the proposed model. A domain will not be included in the domain graph, and the suggested model will not work for it, if it never exchanges IP addresses with any other domains.

## 5. Conclusions

In this research, we provide a novel approach for identifying malicious domains by means of passively examining DNS data. This technique makes use of the fact that malicious domains tend to change over time in order to find robust correlations between them. These associations are then used to infer new malicious domains from a pool of previously discovered malicious domains. The main discovery is that machine learning may ameliorate the shortcomings of blacklists, heavyweight and lightweight classifiers for identifying malicious URLs. Using a genetic approach for feature selection (optimal number of features) and the two-step QABC method, a real-time, accurate, middleweight, and quick classification model was created. Using an ideal number of attributes (middleweight) taken from the initial feature vector has apparent benefits over bigger feature sets. The study built a model for real-time feature collection, and the verified supervised classification and bio-inspired feature selection may work together to develop a stronger classifier.

This paper explored a hybrid machine learning approach (K-mean and QABC) that employed chosen discriminative features and Hadoop to handle large URL data sets. Experimental results demonstrate that the recommended model can achieve high true positive rates and low false positive rates with excellent expansion, i.e., detecting a large group of possibly malicious domains with a small number of seeds. This experiment shows the model’s real-world viability. In this case study, the Hadoop framework examined 10 million domains in about an hour. The results showed that the recommended model could scale to 10 million query-answer pairs with 96.6% accuracy. The following are future plans: (1) application of this method to cyber data types outside DNS for different passive eavesdropper and stealthy attacks [74,75,76,77]; (2) improvement of feature extraction and representation learning (e.g., via deep learning approaches); (3) improvement of GPU web page classification speed.

## Figures and Tables

**Figure 1 biomimetics-08-00197-f001:**
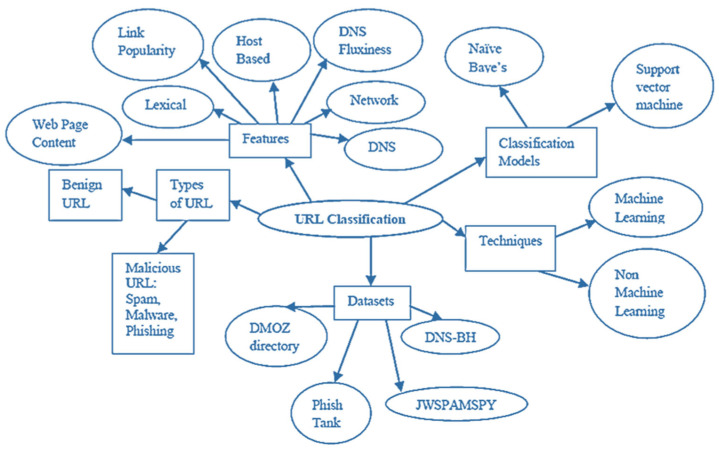
URL Classification [26].

**Figure 2 biomimetics-08-00197-f002:**
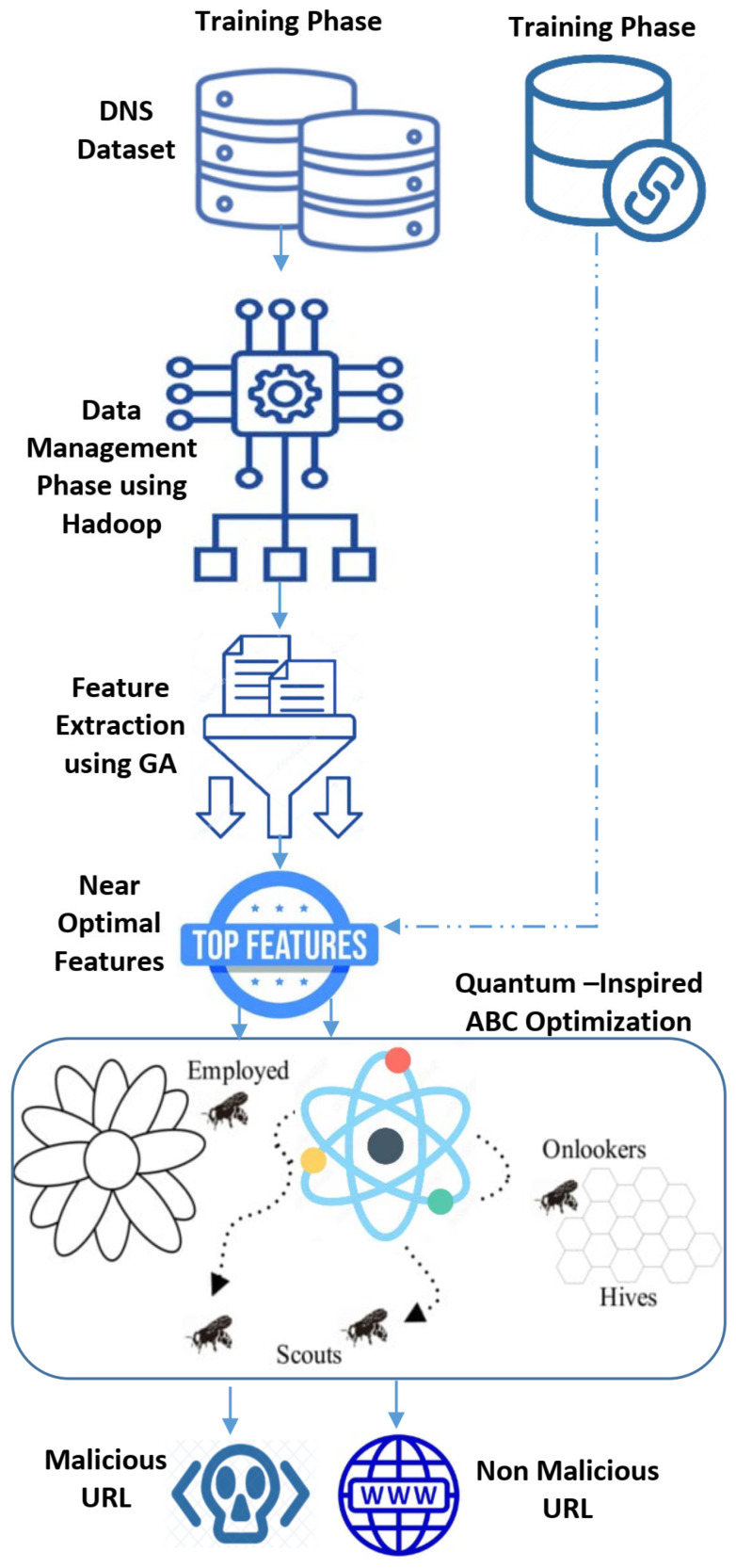
The Proposed Quantum –Inspired Malicious URL Detection Model.

**Figure 3 biomimetics-08-00197-f003:**
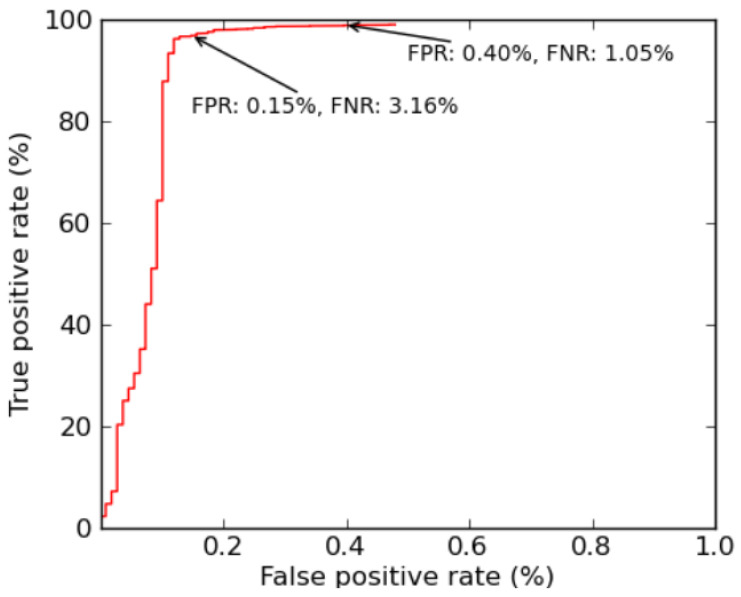
ROC Graph Showing Trade-off between False Negatives and False Positives.

**Table 1 biomimetics-08-00197-t001:** A Record from the used features vector.

Feature	Feature Type	Description	Value
pkt.sniff_timestamp	Time	The time used to sniff sending packet by sniffer software	1466542358.83808 (Microseconds)
qry_class	Type 1	Query class, 1 is most common for Internet	1
qry_name	Type 3	Query Name	www.google.com, accessed on 1 May 2022
qry_name_len	Type 3	Represents the length in bytes of the query name.	123
qry_type	Type 1	Type of DNS packet requested	28
resp_class	Type 1	This defines the protocol family for the Resource Records record	1
resp_len	Type 1	Resource record length	16
resp_name	Type 1	Resource record query name	www.google.com, accessed on 1 May 2022
resp_ttl	Type 2	Resource record time to live in seconds- how long the resource records may be cached	141
resp_type	Type 1	Resource record type	28
Id			54,323
flags_opcode	Type 1	Type of query (e.g., 0 = standard query)	0
flags_response	Type 1	Response code (e.g., 0 = DNS Query completed successfully	1
Aa	Type 1	Authoritative answer (0 = This server isn’t an authority for the domain name or from cache)	Null
count_queries	Type 1	Question count	1
count_auth_rr	Type 1	Authority resource record count	0
count_answers	Type 1	Answer resource record count	1

**Table 2 biomimetics-08-00197-t002:** Confusion Matrix for Optimal and all Features (Average Percentage).

	*TP*%	*FP*%	*FN*%	*TN*%	Testing Phase’s Running Time for 1000 URLs (Milliseconds)
Utilizing all Features	96	6	4	94	2300
Utilizing optimized Features	98	3	2	97	760

**Table 3 biomimetics-08-00197-t003:** Classification Results for Different Classifiers (Average Percentage).

Classifier	Accuracy	*TP* Rate	*TN* Rate	Testing Phase’s Running Time for 1000 URLs (Milliseconds)
Random Forest	95	96	92	500
C4.5	93	92	94	300
SVM	87	88	93	560
ABC	94	96	91	650
Two-step ABC [25]	96	98	95	690
QABC(Proposed Model)	98	98	97	760

**Table 4 biomimetics-08-00197-t004:** Error Rates of the Suggested Classifier after Training them on Different Batches.

Batch Number	Error Rate
batch 1	2.30
batch 2	2.25
batch 3	2.28
batch 4	2.20
batch 5	2.10
batch 6	2.00
batch 7	2.29
batch 8	2.05
batch 9	2.24
batch 10	2.26

**Table 5 biomimetics-08-00197-t005:** Comparative Results for Different Metaheuristics-Based Feature Selection Algorithms.

Feature Selection Module	Accuracy (%)	Feature Reduction	Testing Phase’s Running Time for 1000 URLs (Milliseconds)
BGW	78.52	17%	1230
BB	77.35	39%	940
ACO	98.00	40%	925
GA (Proposed model)	97.50	50%	780

## Data Availability

Datasets for this research are available at https://hackernoon.com/ (accessed on 1 May 2022).

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
