# Peer review of "Building an Effective Classifier for Phishing Web Pages Detection: A Quantum-Inspired Biomimetic Paradigm Suitable for Big Data Analytics of Cyber Attacks"

_biomimetics, 2023, doi:10.3390/biomimetics8020197_

Round 1
Reviewer 1 Report
This paper proposes a model for finding malicious domains by passively analyzing DNS data, which is of practical importance. Using a genetic approach for feature selection and the two-step QABC method, a classification model was created. The following comments are recommended to improve the quality of the paper.
(1) The authors should explain “URL” in formula (7) when you first use this abbreviation.
(2) The size of Figure 1 and Figure 2 should be adjusted.
(3) In section 4, is it possible to highlight the superiority of the proposed approach in the actual physical platform?
(4) Please add more papers related to this work in the references to emphasize the contributions of this work. For example, “The vulnerability of cyber-physical system under stealthy attacks. IEEE Transactions on Automatic Control”.
(5) Proofread the paper again to correct typographical errors.
Author Response
To The Editor-in-chief and the Associate Editor,
Biomimetics.
|
Title: Building an Effective Classifier for Phishing Web Pages Detection: A Quantum-Inspired Biomimetic Paradigm Suitable in Big Data Analytics of Cyber Attacks
Dear Sir,
Thanks for giving us the opportunity to revise the paper for the possible publication in your interesting journal. We also thank the reviewers for their constructive comments. We have significantly revised the manuscripts based the comments received and the details are summarized below.
Reply to the comments:
In the revised version, what was modified and added was written in red
Reviewer 1
|
S. No |
Comments |
Changes made in the revised manuscript based on the comments |
|
1 |
The authors should explain “URL” in formula (7) when you first use this abbreviation. |
In the revised version, it was confirmed that all variables in the equations were defined for the first time. |
|
2 |
The size of Figure 1 and Figure 2 should be adjusted. |
In the revised version, the alignment has been modified for all figures.
In general, the editor will send the paper after acceptance for proofreading, in which the figures and tables will be formatted according to the journals style. |
|
3 |
In section 4, is it possible to highlight the superiority of the proposed approach in the actual physical platform? |
In the revised version, a new paragraph was added at the beginning of the methodology section to highlight the superiority of the proposed approach on the actual physical platform (e.g., firewall).
“Firewalls, key components for secured network infrastructures, are faced with two different kinds of challenges: first, they must be fast enough to classify network packets at line speed, and second, their packet processing capabilities should be versatile in order to support complex filtering policies. Unfortunately, most existing classification systems do not qualify equally well for both requirements; systems built on spe-cial-purpose hardware are fast but limited in their filtering functionality. In contrast, software filters provide powerful matching semantics but struggle to meet line speed. The key challenge in such a design arises from the dependencies between classification rules due to their relative priorities within the rule set: complex rules requiring soft-ware-based processing may be interleaved at arbitrary positions between those where hardware processing is feasible [4][6][8][46]. The superiority of the proposed approach on the physical platform (Firewall) lies in the fact that it is effective against modern attacks, cannot be circumvented with spoofing, cannot make decisions based on ap-plication or authentication, and allows for a huge, easily managed rule list”.
|
|
4 |
Please add more papers related to this work in the references to emphasize the contributions of this work. For example, “The vulnerability of cyber-physical system under stealthy attacks. IEEE Transactions on Automatic Control”. |
In the revised version, more references were added to the reference list (Refs.[74-77])
74. T. Sui, Y. Mo, D. Marelli, X. Sun, and M. Fu,” The vulnerability of cyber-physical system under stealthy attacks”, IEEE Transactions on Automatic Control , Vol. 66, No. 2, pp. 637-50, 2020 75. T. Sui, and X. Sun, “The vulnerability of distributed state estimator under stealthy attacks”, Automatica. Vol. 133, 109869, pp.1-12, 2021. 76. T. Sui, D. Marelli, X. Sun, and M. Fu, “ Stealthiness of Attacks and Vulnerability of Stochastic Linear Systems”, In Proceedings of the IEEE Asian Control Conference, pp. 734-739, 2019. 77. T. Sui, D. Marelli, X. Sun, and K. You, “ A networked state estimation approach immune to passive eavesdropper”, In Proceedings of the Chinese Control Conference, pp. 8867-8869, 2019.
The work presented in this research is inspired by a research plan for the second researcher, which represents a requirement for obtaining a master's degree and includes dealing with DNS data only. In the future work section, we added a sentence in which the suggested version can be extended to deal with passive eavesdroppers and stealthy attacks.
“Applying this method to cyber data types outside DNS for different passive eavesdropper and stealthy attacks [66-69] “
|
|
5 |
Proofread the paper again to correct typographical errors. |
In the revised version, the manuscript was proofread for any grammar mistakes and to correct typographical errors using the grammar checker software (commercial software). https://quillbot.com/grammar-check |

Reviewer 2 Report
The paper describes new research proposing a model for detecting malicious domains using DNS data. The proposed model utilizes a genetic algorithm for selecting DNS data features and a two-step Quantum Ant Colony Optimization (QABC) algorithm for classification.
The paper seems solid, novel, and well-written. The authors demonstrate that their proposal can achieve a reasonably good performance (over 96%) under the chosen dataset (public).
After reading the manuscript, I recommend the acceptance of this paper.
Author Response
To The Editor-in-chief and the Associate Editor,
Biomimetics.
|
Title: Building an Effective Classifier for Phishing Web Pages Detection: A Quantum-Inspired Biomimetic Paradigm Suitable in Big Data Analytics of Cyber Attacks
Dear Sir,
Thanks for giving us the opportunity to revise the paper for the possible publication in your interesting journal. We also thank the reviewers for their constructive comments. We have significantly revised the manuscripts based the comments received and the details are summarized below.
Reply to the comments:
In the revised version, what was modified and added was written in red
Reviewer 2
|
S. No |
Comments |
Changes made in the revised manuscript based on the comments |
|
1 |
The paper describes new research proposing a model for detecting malicious domains using DNS data. The proposed model utilizes a genetic algorithm for selecting DNS data features and a two-step Quantum Ant Colony Optimization (QABC) algorithm for classification.
The paper seems solid, novel, and well-written. The authors demonstrate that their proposal can achieve a reasonably good performance (over 96%) under the chosen dataset (public).
After reading the manuscript, I recommend the acceptance of this paper. |
Thank you for positive feedback |

Reviewer 3 Report
This paper presents new research that proposes a model for finding malicious domains by passively analyzing DNS data. In detail, this paper proposes a real-time, accurate, middleweight, and fast classifier by combining a genetic algorithm for selecting DNS data features with a two-step Quantum Ant Colony Optimization (QABC) algorithm for classification. Hereafter, you can find my comments:
Line 271] May you clarify how you extracted the dataset?
Lines 279-281] May you clarify if you removed all IP addresses from the dataset?
Lines 283-299] May you clarify which kind of pre-processing you performed?
Line 328] What do you mean by "Domain-Name-based Features"?
Line 361 Which are the 17 base features?
Step 4] The usage of Genetic Algorithms is not clear at all. Please add more details or even a diagram to explain how you used it for feature selection. The final selected features are neither specified.
Step 5] You added lot of formalism about QABC but it is totally not clear how this method is applied to malicious URL detection.
Section 4.1] You have not specified the total vs selected number of features.
Section 4.2] I see minimal difference between classifier. Did you evaluate also computational complexity? Are you sure other classifiers, e.g., SVM, are not simpler than yours?
Section 4.4] How are the batches selected to demonstrate the concept drift?
The paper discusses an interesting topic, but it is not clear how the proposed techniques are effectively working to detect malicious URLs. Some key aspects, such as the used features, have been covered very high level. The addition of some examples may have clarified a lot.
Author Response
To The Editor-in-chief and the Associate Editor,
Biomimetics.
|
Title: Building an Effective Classifier for Phishing Web Pages Detection: A Quantum-Inspired Biomimetic Paradigm Suitable in Big Data Analytics of Cyber Attacks
Dear Sir,
Thanks for giving us the opportunity to revise the paper for the possible publication in your interesting journal. We also thank the reviewers for their constructive comments. We have significantly revised the manuscripts based the comments received and the details are summarized below.
Reply to the comments:
In the revised version, what was modified and added was written in red
Reviewer 3
|
S. No |
Comments |
Changes made in the revised manuscript based on the comments |
|
1 |
Line [271] May you clarify how you extracted the dataset? |
The experiments are conducted using a benchmark dataset from https://www.circl.lu/services/passive-dns/ website that are comprised of captured passive DNS data, which are answers (e.g., IP address, time to live (TTL), record counts) from authoritative DNS servers given domain name queries from the browsers of users. (Page 6)
In the revised version, more sentences were added to highlight how were the dataset obtained.
“CIRCL Passive DNS is a database storing historical DNS records from various resources including malware analysis or partners. The DNS historical data is indexed, which makes it searchable for incident handlers, security analysts or researchers”. (Page 13)
|
|
2 |
Lines [279-281] May you clarify if you removed all IP addresses from the dataset? |
In the revised version, I new sentence was added to clarify that IP addresses were removed from the dataset as the anonymization was considered in our case.
“The authors in [37] employed largely IP-based computations, but because our dataset was anonymized, this was not feasible, so IP addresses were removed from the dataset. (page 7) |
|
3 |
Lines [283-299] May you clarify which kind of pre-processing you performed? |
In the revised version, a new paragraph was added to clarify the preprocessing step that was employed in the suggested model
“Data cleaning is the basic phase of data preparation, identifying defective data and incorrect information. The cleaning stage modifies or deletes incomplete, erroneous, inconsistent, and irrelevant data. Data cleaning can be performed on Hadoop projects using Apache Hive. Hive supports custom user-defined functions (UDF) for tasks like data cleansing and filtering. Hive UDFs can be defined according to programmers' requirements [51]. Hive operates on the cluster's server-side [52].” (Page 7) |
|
4 |
Line [328] What do you mean by "Domain-Name-based Features"? |
In the revised version, a new set of sentences was added to clarify the meaning of domain-name-based features.
“This feature represents QNAME query name in which the target domain-name is presented as a sequence of labels, each label consisting of a length octet followed by that number of octets. The domain name ends with the zero length octet for the null label of the root. This DNS server looks for Resource Records (RRs) that match the specified Qtype and Qclass. If it does not have a match, this server can point to a DNS server that may have a match instead. QTYPE A two octet code which specifies the type of the query (e.g., host addresses). QCLASS A two octet code that specifies the class of the query (e.g., Internet addresses”.
(Page 8) |
|
5 |
Line [361] Which are the 17 base features? |
In the revised version, a new table was added to clarify the 17 feature vectors used for each URL before selecting optimal features based on a genetic algorithm Table 1, Page 8 |
|
6 |
Step [4] The usage of Genetic Algorithms is not clear at all. Please add more details or even a diagram to explain how you used it for feature selection. The final selected features are neither specified. |
In the revised version, two algorithms were added to give more details on how to use GA for feature selection.
Algorithm 1: Feature Selection Algorithm Algorithm 2: Genetic Algorithm Pseudo Code (Page 10)
Furthermore, a new sentence was added to clarify the length of the final selected features.
Based on the results, a 10-element feature vector is optimal for use, allowing the proposed classifier to achieve the lowest error rate. Remember that the GA's non-deterministic nature implies that the optimal feature vector's elements may vary with each program run.
(Page 11) |
|
7 |
Step [5] You added lot of formalism about QABC but it is totally not clear how this method is applied to malicious URL detection. |
In the revised version, a new algorithm (Algorithm 5) was added that summarize how the suggested model was utilized malicious URL detection.
(Algorithm 5, page 14) |
|
8 |
Section [4.1] You have not specified the total vs selected number of features. |
In the revised version, a new sentence was added to specify the ratio of the total number of employed features to the total number of optimized features extracted by employing the genetic algorithm procedure.
“In general, the use of the genetic algorithm results in the reduction of the feature vector size 1×17, to a feature vector of size 1× 10. So the number of features (optimized features) used for each URL was reduced to 58% of the total number of features (all features).”
(Page 11) |
|
9 |
Section [4.2] I see minimal difference between classifier. Did you evaluate also computational complexity? Are you sure other classifiers, e.g., SVM, are not simpler than yours? |
In the revised version, a new set of experiments was conducted to highlight the role of different metaheuristics-based feature extraction algorithms in terms of accuracy as compared with the employed GA-based feature selection module.
(Table 4, page 16)
Experiment 5: (Performance of different metaheuristic –based feature selection algorithms) “Feature selection is the process of selecting the best feature among a huge number of features in a dataset. However, the problem in feature selection is to select a subset that performs better under some classifier. Feature selection is frequently used in machine learning for processing high-dimensional data. It reduces the number of features in the dataset and makes the classification process easier. This procedure can not only reduce irrelevant features but, in some cases, increase classification performance due to the finite sample size. Meta-heuristic algorithms are widely adopted for feature selection due to their enhanced searching ability. Though current state-of-the-art systems have demonstrated impressive performance, there is still no consensus on the optimum feature selection algorithm for the task of phishing web page detection. In this experiment, we investigate the performance of three metaheuristic feature selection algorithms: ant colony optimization (ACO), binary bat (BB), and binary grey wolf (BGW). In this case, we swap between different feature selection modules in step 4 and compare the results to those obtained by the genetic algorithm (GA) that is utilized by the proposed model for feature selections for the same task. For implementation, each module was embedded into the main model as a blackbox with its default parameters. See [67–73] for more details about the mentioned metaheuristic optimization techniques and their default parameters. As revealed from Table 4, both GA and ACO have gained better classification accuracy than other algorithms. However, utilizing GA achieves 50% feature reduction as compared with ACO, which achieves only 40%. There is no significant difference between them in the accuracy of the classification, but the use of GA for feature selection reduces the computational complexity by 20% on average. In general, one of the factors that influences the increment of time and iteration is the population being initialized. The way ACO initializes the population by using state transition rules is more efficient compared to GA, which is based on random approaches.
Furthermore, a new column was added for both Table 1 and Table 2 to highlight the testing phase’s running time for batches of URLs in milliseconds as an indicator of the computational complexity analysis of the proposed model and the comparative classifiers. (page 13&14)
“The suggested model takes more time to classify URLs than competing approaches, as indicated in Table 2, but it also provides higher accuracy.” |
|
10 |
Section [4.4] How are the batches selected to demonstrate the concept drift? |
In the revised version, a new sentence was added to illustrate how the batches were selected.
“The passive DNS data collection is separated randomly into 10 batches, each comprising 1500000 benign and malicious URLs. In our case, OperateBatchDomain function was used to submit a task for adding domain names or DNS records in batches”.
(Page 17) |
|
11 |
The paper discusses an interesting topic, but it is not clear how the proposed techniques are effectively working to detect malicious URLs. Some key aspects, such as the used features, have been covered very high level. The addition of some examples may have clarified a lot. |
Thank you for positive feedback In the revised version, a new experiment was added to clarify how the suggested model is effective in detecting malicious URLs. 4.5. Experiment 5: (Performance of different metaheuristics –based feature selection algorithms) (Page 18) Furthermore, many algorithms were added to clarify how the suggested model works to detect malicious URLs. (Algorithms 1,2,3, and 5) (Pages 10,12, and 14) Furthermore, many sentences were added clarify the used and optimize URL’ features (Page 11) |

Reviewer 4 Report
The authors investigate the problem of malicious domains through the passive analysis of DNS data.
They propose a combined framework based on a genetic algorithm along with a quantum-based Ant Colony Optimization algorithm for feature selection.
The experimental part is based on the Hadoop framework useful to prove the effectiveness of the proposed method.
Finally, performance of proposed method is contrasted with several standard algorithms including Random Forest, C4.5, SVM, simple ABC.
The paper seems to be interesting. This notwithstanding, I have a major concern.
Actually, the authors just mention classic performance indicators (e.g. Accuracy, TP Rate, TN rate, etc.) to compare the algorithms, but nothing about computational complexity seems to be afforded.
When dealing with machine learning-based techniques in the field of cybersecurity, it is crucial to take into account the time needed to perform security checks. In many cases, in fact, cyber attacks can be faster than defense mechanisms, and some feature selection algorithms such as Ant colony can require some time to be employed.
I suggest to analyze and/or mention some papers where FS/genetic algorithms are analyzed both in terms of performance and required time. Some hints follow:
- Supervised Feature Selection Techniques in Network Intrusion Detection: a Critical Review (Engineering Application of Artificial Intelligence, 2021)
- Optimizing Uncertain Express Delivery Path Planning Problems with Time Window by Ant Colony Optimization (Proc. Of IEEE CIS, 2021)
- A multi-objective ant colony algorithm for the optimization of path planning problem with time window (Proc. Of IEEE CIS, 2022)
Author Response
To The Editor-in-chief and the Associate Editor,
Biomimetics.
|
Title: Building an Effective Classifier for Phishing Web Pages Detection: A Quantum-Inspired Biomimetic Paradigm Suitable in Big Data Analytics of Cyber Attacks
Dear Sir,
Thanks for giving us the opportunity to revise the paper for the possible publication in your interesting journal. We also thank the reviewers for their constructive comments. We have significantly revised the manuscripts based the comments received and the details are summarized below.
Reply to the comments:
In the revised version, what was modified and added was written in red
Reviewer 4
|
S. No |
Comments |
Changes made in the revised manuscript based on the comments |
|
1 |
The paper seems to be interesting. |
Thank you for positive feedback |
|
2 |
Actually, the authors just mention classic performance indicators (e.g. Accuracy, TP Rate, TN rate, etc.) to compare the algorithms, but nothing about computational complexity seems to be afforded. |
In the revised version, a new paragraph was added in Section 4 to illustrate the difference between complexity analysis and running time analysis. “In computer science, the analysis of algorithms is the process of finding the computational complexity of algorithms—the amount of time, storage, or other resources needed to execute them. Usually, this involves determining a function that relates the size of an algorithm's input to the number of steps it takes (its time complexity) or the number of storage locations it uses (its space complexity). Time complexity is commonly estimated by counting the number of elementary operations performed by the algorithm, where an elementary operation takes a fixed amount of time to perform. As the prototype of the proposed model was built using off-the-shelf software that contains many functions that call each other, it becomes theoretically difficult to calculate time and space complexities. Time complexity is a complete theoretical concept related to algorithms, while running time is the time a code would take to run. Run-time analysis was utilized as an indicator of the different approaches’ computational complexity. An algorithm is said to be efficient when this function's values are small or grow slowly compared to the size of the input [66]. On average, the proposed technique re-quires around one-third as much time to run in the testing phase as it does in the training phase to accomplish classification.”
Furthermore, a new column was added for both Table 1 and Table 2 to highlight the testing phase’s running time for batches of URLs in milliseconds as an indicator of the computational complexity analysis of the proposed model and the comparative classifiers. (page 13&14)
“The suggested model takes more time to classify URLs than competing approaches, as indicated in Table 2, but it also provides higher accuracy.”
|
|
3 |
When dealing with machine learning-based techniques in the field of cybersecurity, it is crucial to take into account the time needed to perform security checks. In many cases, in fact, cyber-attacks can be faster than defense mechanisms, and some feature selection algorithms such as Ant colony can require some time to be employed. I suggest to analyze and/or mention some papers where FS/genetic algorithms are analyzed both in terms of performance and required time. Some hints follow:
- Supervised Feature Selection Techniques in Network Intrusion Detection: a Critical Review (Engineering Application of Artificial Intelligence, 2021) - Optimizing Uncertain Express Delivery Path Planning Problems with Time Window by Ant Colony Optimization (Proc. Of IEEE CIS, 2021) - A multi-objective ant colony algorithm for the optimization of path planning problem with time window (Proc. Of IEEE CIS, 2022)
|
In the revised version, a new set of experiments was conducted to highlight the role of different metaheuristics-based feature extraction algorithms in terms of accuracy as compared with the employed GA-based feature selection module.
(Table 4, page 16)
Experiment 5: (Performance of different metaheuristic –based feature selection algorithms) “Feature selection is the process of selecting the best feature among a huge number of features in a dataset. However, the problem in feature selection is to select a subset that performs better under some classifier. Feature selection is frequently used in machine learning for processing high-dimensional data. It reduces the number of features in the dataset and makes the classification process easier. This procedure can not only reduce irrelevant features but, in some cases, increase classification performance due to the finite sample size. Meta-heuristic algorithms are widely adopted for feature selection due to their enhanced searching ability. Though current state-of-the-art systems have demonstrated impressive performance, there is still no consensus on the optimum feature selection algorithm for the task of phishing web page detection. In this experiment, we investigate the performance of three metaheuristic feature selection algorithms: ant colony optimization (ACO), binary bat (BB), and binary grey wolf (BGW). In this case, we swap between different feature selection modules in step 4 and compare the results to those obtained by the genetic algorithm (GA) that is utilized by the proposed model for feature selections for the same task. For implementation, each module was embedded into the main model as a blackbox with its default parameters. See [67–73] for more details about the mentioned metaheuristic optimization techniques and their default parameters. As revealed from Table 4, both GA and ACO have gained better classification accuracy than other algorithms. However, utilizing GA achieves 50% feature reduction as compared with ACO, which achieves only 40%. There is no significant difference between them in the accuracy of the classification, but the use of GA for feature selection reduces the computational complexity by 20% on average. In general, one of the factors that influences the increment of time and iteration is the population being initialized. The way ACO initializes the population by using state transition rules is more efficient compared to GA, which is based on random approaches.
Furthermore, new references were added to the references list Refs. [67–73] that were used to justify the results.
67. Thakkar, A., Lohiya, R. Attack classification using feature selection techniques: a comparative study. Journal of Ambient Intelligence and Humanized Computing, vol. 12, pp. 1249-1266, 2021. 68. Hassani, H., Hallaji, E., Razavi-Far, R., Saif, M. Unsupervised concrete feature selection based on mutual information for diagnosing faults and cyber-attacks in power systems. Engineering Applications of Artificial Intelligence, vol. 100, 104150, pp.1-13, 2021. 69. Bouzoubaa, K., Taher, Y., Nsiri, B. Predicting DOS-DDOS attacks: Review and evaluation study of feature selection methods based on wrapper process. International Journal of Advanced Computer Science and Applications, vol. 12, no. 5, pp. 131-145, 2021. 70. Garg, S., Verma, S. A Comparative Study of Evolutionary Methods for Feature Selection in Sentiment Analysis. In Proceedings of the International Joint Conference on Computational Intelligence, pp. 131-138, 2019. 71. Di Mauro, M., Galatro, G., Fortino, G., Liotta, A. Supervised feature selection techniques in network intrusion detection: A critical review. Engineering Applications of Artificial Intelligence, vol. 101, 104216, pp.1-15, 2021. 72. Yi, Y., Wang, Y., Gu, F., Chen, X. Optimizing uncertain express delivery path planning problems with time window by ant colony optimization. In Proceedings of the International Conference on Computational Intelligence and Security, pp. 420-424, 2021. 73. Deng, C., Lin, J., Chen, L. A multi-objective ant colony algorithm for the optimization of path planning problem with time window. In Proceedings of the International Conference on Computational Intelligence and Security, pp. 351-355, 2022.
|

Round 2
Reviewer 3 Report
Thanks for the manuscript updates to clarify the aspects I asked for. I believe it can be published in the present form.
Reviewer 4 Report
In this revised version, the authors made a good effort to address all my comments raised in the first round of review. In particular, they have:
- Added in Sect. 4 new information about time complexity/run-time analysis;
- Clarified the role of the adopted feature selection techniques;
- Improved the related work section.
In my opinion, the manuscript can be accepted in its current form.